# Potential Risk Factors to COVID-19 Severity: Comparison of SARS-CoV-2 Delta- and Omicron-Dominant Periods

**DOI:** 10.3390/ijerph21030322

**Published:** 2024-03-10

**Authors:** Daiki Yamaguchi, Odgerel Chimed-Ochir, Yui Yumiya, Eisaku Kishita, Tomoyuki Akita, Junko Tanaka, Tatsuhiko Kubo

**Affiliations:** 1Department of Public Health and Health Policy, Graduate School of Biomedical and Health Sciences, Hiroshima University, Hiroshima 734-8553, Japan; b191365@hiroshima-u.ac.jp (D.Y.); yumiya@hiroshima-u.ac.jp (Y.Y.); tkubo@hiroshima-u.ac.jp (T.K.); 2Medical Economics Division, Health Insurance Bureau, Ministry of Health, Labour and Welfare, Tokyo 100-8916, Japan; kishita-eisaku@mhlw.go.jp; 3Department of Epidemiology, Infectious Disease Control and Prevention, Graduate School of Biomedical & Health Sciences, Hiroshima University, Hiroshima 734-8553, Japan; tomo-akita@hiroshima-u.ac.jp; 4Medical Policy Office, Hiroshima University, Hiroshima 734-8553, Japan; jun-tanaka@hiroshima-u.ac.jp

**Keywords:** COVID-19, risk factors, J-SPEED, Japan, data collection

## Abstract

Background: Continued study of risk factors can inform future pandemic preparedness and response. We aimed to determine the potential risk factors of COVID-19 severity among patients admitted to the hospital during the Delta- and Omicron-dominant periods. Methods: We utilized the J-SPEED-style COVID-19 Hospital version, a pre-administered questionnaire, to collect data from hospitals in Hiroshima Prefecture between 8 August 2021 and 19 April 2022. Results: During the Delta-dominant period, patients aged over 65 (OR = 2.59, 95% CI = 1.75–3.84), males (OR = 1.42, 95% CI = 1.12–1.81) and with BMI exceeding 25 (OR = 1.99, 95% CI = 1.57–2.52), diabetes (OR = 2.03, 95% CI = 1.40–2.95), and those with fewer than two doses of vaccine (OR = 2.39, 95% CI = 1.46–3.91) were at a greater risk of severe COVID-19 compared to those without these risk factors. During the Omicron-dominant period, significantly greater severity was observed among patients over 65 years old (OR = 3.89, 95% CI = 2.95–5.12), males (OR = 1.76, 95% CI = 1.40–2.21), those with high blood pressure (OR = 1.30, 95% CI = 1.02–1.65), and mental disorder (OR = 2.22, 95% CI = 1.69–2.92) compared to patients without these risks. Conclusions: Our findings indicate that risk factors vary across different SARS-CoV-2 variants. Examining variant-specific risk factors for COVID-19 severity can aid policymakers, public health specialists, and clinicians in prioritizing screening, treatment, and vaccination efforts, especially during potential healthcare resource shortages.

## 1. Introduction

The health systems of the countries of the world were detrimentally affected by COVID-19. Since the first case of COVID-19 was identified in Wuhan, China, in December 2019 [1], the original SARS-CoV-2 virus has undergone multiple mutations. In late 2020, the B.1.617 variant (Delta) was first detected in India [2,3], with Japan reporting its first case of the Delta variant on 28 March 2021 [4]. Subsequently, the B.1.1.529 variant (Omicron) was first observed in South Africa on 26 November 2021 [5] and was first detected in Japan on 30 November 2021 [6], overtaking the previously predominating Delta variant. Previous reports have suggested that Omicron infection is less severe but more contagious than earlier variants, including the Delta variant [7,8,9]. Furthermore, several countries have reported lower rates of hospital admission, the need for intensive care and mortality during periods when the Omicron variant was dominant compared to periods when other variants of concern were dominant [8,10]. A prospective observational study showed that the likelihood of hospitalization is 25% lower for the Omicron variant than for the Delta variant [11]. Additionally, numerous efforts have been made to investigate demographic, behavioral and other risk factors for the severity of COVID-19 during periods dominated by either the Delta or Omicron variant [12,13,14]. However, we contend that examining the impact of these factors during both the Delta- and Omicron-dominant periods within specific populations provides valuable insights into how the dynamics of COVID-19 severity may have shifted over time. Moreover, while the general risk factors may be well known, our study offers local data that can inform targeted interventions and public health strategies tailored to specific populations.

While the intensity of the pandemic has abated, ongoing studies on risk factors remain crucial for informing future pandemic preparedness and response efforts. Understanding the risk factors associated with severe COVID-19 during periods dominated by the Delta and Omicron variants serves several vital purposes. Healthcare providers can leverage this knowledge to effectively identify and prioritize high-risk individuals for targeted interventions, such as vaccination campaigns, early treatment, and vigilant monitoring [15,16]. Moreover, comprehending these risk factors aids healthcare providers in making informed decisions regarding patient management, encompassing the judicious use of antiviral therapies, oxygen supplementation, and hospitalization protocols [15,16]. This knowledge is particularly invaluable for clinicians facing potential shortages of healthcare resources, such as hospital beds, enabling them to make prudent clinical decisions [17]. Examining the risk factors that affect COVID-19 severity may also help policymakers in prioritizing screening, other non-pharmaceutical treatments and vaccination [18].

Therefore, we aimed to determine the relationship between various potential risk factors and the severity of COVID-19 among hospitalized patients during the Delta- and Omicron-dominant periods in Hiroshima Prefecture, Japan.

## 2. Materials and Methods

### 2.1. Data Collection

Hiroshima prefecture implemented a unique surveillance system during the COVID-19 pandemic, enabling the timely collection of data from various healthcare facilities including public health centers, recuperation hotels, online treatment centers, oxygen centers, PCR centers, and hospitals [19]. This surveillance was facilitated through the utilization of a standardized data collection form known as J-SPEED (Japanese Surveillance in Post-Extreme Emergencies and Disasters). The J-SPEED-style COVID-19 forms were developed based on the lessons learned from the 2011 Great East Japan Earthquake, with the concept of simplifying and standardizing health data collection to collect data in near-real time. In our study, we specifically focused on analyzing the J-SPEED COVID-19 Hospital version, which was initiated in Hiroshima Prefecture, Japan, in May 2020. Hospitals within Hiroshima Prefecture that offer inpatient care for COVID-19 patients are encouraged to utilize this form to record pertinent patient information. However, out of the 43 hospitals in Hiroshima Prefecture, only 21 currently utilize this form for patient data registration. Nonetheless, from 8 April 2021 to 19 April 2022, a total of 5807 COVID-19 patients’ data were collected using this form, representing 55% of the total patients admitted to hospitals in Hiroshima Prefecture during the designated period.

The J-SPEED COVID-19 Hospital version encompasses 54 items, covering a wide range of information including demographic details, admission method, lifestyle risk factors such as smoking and a BMI, vaccination status, severity of illness, symptoms, existing diseases, treatment modalities, and outcomes.

### 2.2. Data Analysis

We extracted the data collected during the period (from 1 August to 30 November 2021) in which the Delta variant was dominant (hereinafter the Delta-dominant period) and the period (from 22 December 2021 to 19 April 2022) in which the Omicron variant of SARS-CoV-2 was dominant (hereinafter the Omicron-dominant period).

The risk factors examined in the study included being over the age of 65, being male, being a current smoker, having a BMI over 25, currently under hemodialysis, high blood pressure, diabetes mellitus type 2, and dementia and having received fewer than two doses of COVID-19 vaccination. These risk factors were selected in accordance with the guideline for the treatment of COVID-19 of the Ministry of Health Labor and Welfare of Japan [20].

Severity in our study was categorized into four levels: very severe (requiring extracorporeal membrane oxygenation—ECMO), severe (requiring artificial respiration), moderate (requiring oxygen), and mild (no oxygen required). Due to an insufficient number of cases in the very severe and severe categories, we combined them with the moderate cases, forming a single category termed “more than moderate” severe cases for analysis purposes.

The severity rate was then calculated by dividing the number of “more than moderate” severe cases by the total number of patients in the corresponding periods, and this was expressed as a percentage.

First, we conducted a descriptive analysis of the risk factors among patients and the severity of COVID-19 during both the Delta- and Omicron-dominant periods. Second, we visually depicted the admission patterns due to COVID-19 during both the Delta- and Omicron-dominant periods, alongside the proportion of “more than moderate” severe cases. Third, logistic regression was employed to assess the association between risk factors and severity. We reported adjusted odds ratios. In logistic regression, the explanatory modeling method was used in the analysis because the goal of the current study is to identify risk factors that are causally related to a COVID-19 infection [21]. We selected the candidate variables [21] including sex, age, life style risk factors, vaccine status, and pre-existing diseases, based on the guidelines for the treatment of COVID-19 of the Ministry of Health Labor and Welfare of Japan [20]. Therefore, we included all candidate variables (age, sex, smoking status, overweight and obesity, vaccine status, blood pressure, diabetes, hemodialysis, mental disorder) into the initial model. The evaluation of risk factors, notably pre-existing conditions such as diabetes and hypertension, was conducted by attending physicians and documented when treatment was warranted. The diagnosis of these conditions was based on clinical judgment and recorded during patient care. Occasional increases in blood pressure or high blood glucose levels that did not meet diagnostic criteria for hypertension or diabetes were not considered as such in our analysis. Subsequently, in alternative model, potential confounder variables, such as types of medicine and existing symptoms, were incorporated into candidate variables to confirm whether these confounders change the relationship between risk factors and the severity of COVID-19, regardless of statistical significance. Following this, we compared the two models, and the model with a lower AIC score was deemed the better-fit model and selected as the final model [21]. In the final model, we assessed the potential multicollinearity problem, which may affect the efficacy of the model, by examining the phi coefficient between all independent variables. The phi coefficient, commonly employed to identify multicollinearity problem in logistic regression [22], ranged from 0.01 to 0.42. This indicates the absence of a multicollinearity problem in the modeling [23]; thus, we proceeded with the following model in our analysis:logit(S) = ln S/(1 − S) = a + b1 × 1 + b2 × 2 + b3 × 3 + b4 × 4 + b5 × 5 + b6 × 6 + b7 × 7 + b8 × 8 + b9 × 9
where S is a probability of being “more than moderate” severe, a is an intercept, ×1–×9 are being older than 65 years old, male, being current smoker, having BMI over 25, hemodialysis, high blood pressure, diabetes, and dementia, and being vaccinated with fewer than 2 doses, respectively. SAS Version 9.4 (SAS Institute, Inc., Cary, NC, USA) was used for data analysis.

Approval for ethical review was obtained from the Ethical Committee of Hiroshima University on 29 June 2021 (approval number: E-2508).

This study was funded by the AMED (grant No.: 21fk0108550h0001).

## 3. Results

Table 1 presents the proportion of severe COVID-19 cases based on various risk factors during the Delta- and Omicron-dominant periods. During the Delta- and Omicron-dominant period of COVID-19 infection, 1367 and 1790 patients were admitted to the hospital, respectively. In total, 42 percent (*n* = 576) of the total patients admitted to the hospital during the Delta-dominant period had severe cases, while 30 percent of patients (*n* = 542) admitted during the Omicron-dominant period had severe cases. Among the patients aged over 65, 54% during the Delta-dominant period and 43% during the Omicron-dominant period had severe cases. Regarding gender, 46% of male patients during the Delta-dominant period and 34% during the Omicron-dominant period experienced severe cases, while for female, non-pregnant patients, the rates were 37% and 28%, respectively.

Patients admitted during the Delta-dominant period exhibited a higher severity rate than those admitted during the Omicron-dominant period across all studied risk factors except for dementia/mental disorder. The number of risks also influenced severity, with higher numbers of risks associated with higher severity rates in both periods. For instance, during the Delta-dominant period, patients with more than four risk factors had a severity rate of 56%, while during the Omicron-dominant period, they had a severity rate of 46%. Conversely, patients with fewer risk factors had lower severity rates. For example, during the Delta-dominant period, patients with one or zero risk factors had a severity rate of 33%, whereas during the Omicron-dominant period, they had a severity rate of 13%.

Figure 1 illustrates the trend of patients admitted to hospitals in Hiroshima Prefecture, alongside the severity of hospitalized patients. During the Delta-dominant period, the severity rate exhibited an initial high level, which continued to rise in conjunction with the increasing number of patients. In contrast, during the Omicron-dominant period, the severity rate commenced at a low level, gradually escalating as the number of patients increased.

Table 2 shows the association between the risk factors studied and the severity rate of COVID-19 infections during the Delta- and Omicron-dominant periods. In the Delta-dominant period, patients over the age of 65 (OR = 2.59, 95% CI = 1.75–3.84, *p* < 0.0001), male (OR = 1.42, 95% CI = 1.12–1.81, *p* = 0.0039) and with a BMI of higher than 25 (OR = 1.99, 95% CI = 1.57–2.52, *p* < 0.0001), diabetes (OR = 2.03, 95% CI = 1.40–2.95, *p* = 0.0002), and fewer than two doses of vaccine (OR = 2.39, 95% CI = 1.46–3.91, *p* = 0.0005) had significantly greater odds of having severe COVID-19 than those who did not have these risk factors.

During the Omicron-dominant period, significantly greater severity was recorded in patients with an age of over 65 (OR = 3.89, 95% CI = 2.95–5.12, *p* < 0.0001), male (OR = 1.76, 95% CI = 1.40–2.21, *p* < 0.0001) high blood pressure (OR = 1.30, 95% CI = 1.02–1.65, *p* = 0.0338), and dementia/mental disorder (OR = 2.22, 95% CI = 2.22, 95% CI = 1.69–2.92, *p* < 0.0001) compared to patients with none of these risks. In addition, when compared to patients with one or no risks, those with two, three, or more than four risks had significantly higher odds of having severe COVID-19 cases during both the Delta- and Omicron-dominant periods.

## 4. Discussion

We used the J-SPEED Hospital form to examine data collected during the Delta-dominant period from 1 August to 30 November 2021, and the Omicron-dominant period from 22 December 2021 to 19 April 2022. Our investigation aimed to compare the risk factors associated with COVID-19 severity between the Delta and Omicron variants. Our findings revealed that being over 65 years old or male were significant risk factors for COVID-19 severity during both the Delta and Omicron variants. Additionally, factors such as overweight and obesity, diabetes, and receiving fewer than two doses of vaccine were also significant risk factors, specifically during the Delta-dominant period, whereas these associations were not observed during the Omicron-dominant period. High blood pressure and dementia emerged as significant risk factors for severity during the Omicron-dominant period.

Our findings align with previous studies that have highlighted older age and male gender as significant risk factors for COVID-19 severity during both the Delta- and Omicron-dominant periods. Regarding the association of gender and the severity of COVID-19, a systematic review has demonstrated that men exhibit a higher likelihood of developing severe COVID-19 disease and are more frequently admitted to intensive care units [17]. Pijls et al. also concluded that men face a heightened risk of severe disease when infected with COVID-19 compared to women. This observation is consistent with the fact that respiratory tract infectious diseases are more severe in men in general [24]. Moreover, differences in the immune systems between men and women may partially account for these disparities in respiratory tract infection severity [25]. Some other groups also have reported that men have a weaker immune response, including the innate immune response [24,26], which is the first mechanism for host defense present in all multicellular organisms [27].

Numerous studies have consistently demonstrated that adults over the age of 65 are at a heightened risk of experiencing severe COVID-19 illness compared to younger individuals. A systematic review highlighted that individuals aged 70 and older who contracted COVID-19 were more prone to severe illnesses and ICU hospitalization compared to those younger than 70 [17]. Similarly, research conducted in South Africa focusing on the Omicron and Delta strains underscored the significantly elevated risks of severe disease among individuals aged 60 and older compared to those aged 19–24 [8]. These findings can be attributed to age-related changes in the immune system. Aging is often associated with a chronic proinflammatory immunological state characterized by persistent low-grade innate immune activation. Such immunological changes may exacerbate tissue damage resulting from infections in older adults [28,29]. Moreover, older individuals infected with COVID-19 are more likely to develop comorbidities and exhibit diminished reserve capacity in vital organs, leading to increased frailty. Coupled with age-related immune system alterations, these factors contribute to poorer outcomes and higher mortality rates among older COVID-19 patients [17].

During the Delta-dominant period, obesity was revealed to be a significant risk factor, whereas it was retained during the Omicron-dominant period. Previous studies have consistently highlighted the association between obesity and poor prognosis in SARS-CoV-2 infection [30,31]. The impact of obesity on poor outcomes is known to vary inversely with age, suggesting that its effect may be attenuated in older individuals while being accentuated in younger individuals [32]. In the current study, it is noteworthy that individuals under the age of 65 comprised 86 percent of all hospitalized patients during the Delta-dominant period. In contrast, this proportion decreased to less than half during the Omicron-dominant period. The substantially smaller number of younger patients hospitalized during the Omicron-dominant period likely contributed to the absence of a significant association between obesity and severity with the Omicron variant.

Regarding smoking, in both the Delta- and Omicron-dominant periods, our analysis did not identify a significant relationship between smoking and COVID-19 severity. The evidence regarding the link between smoking and the severity of COVID-19 is heterogeneous. While some studies have found no association between active smoking and an enhanced risk of developing severe disease [33], others have found smoking as a significant risk factor [34,35]. This discrepancy may be attributed to variations in factors such as the duration of smoking, past smoking history, and the number of cigarettes smoked per day, that were examined across different studies. It is worth mentioning that our study only documented whether patients are current smokers or not, without capturing additional relevant information as aforementioned. This limitation makes it challenging to compare our findings with those of other studies that may have considered a broader range of smoking-related factors.

Regarding high blood pressure, while the majority of existing evidence indicates a significant association between hypertension and severe COVID-19 outcomes [36,37], our finding revealed that high blood pressure was a significant risk factor in the Omicron-dominant period but not in the Delta-dominant period. This discrepancy may be attributed to the intricate relationship between hypertension and age, as older individuals are more likely to have hypertension. Our analysis revealed a smaller proportion of older individuals in the Delta-dominant period compared to the Omicron-dominant period. As hypertension tends to be more prevalent among older age groups, the relatively smaller representation of older individuals in the Delta-dominant period may have contributed to the observed lack of significance in the association between high blood pressure and COVID-19 severity during that period.

Increased morbidity among patients with diabetes mellitus has been observed since the onset of the pandemic [38,39]. However, the underlying reasons for this phenomenon remain a subject of debate [40]. Bassani et al. recently reported findings suggesting that the Omicron variant exhibits a higher affinity between the viral Spike protein and the human receptor ACE2. This led to speculation that diabetic patients may be at an increased risk of experiencing severe COVID-19 with Omicron variants [40]. Despite the proposed mechanism suggesting a potential heightened risk for diabetic patients with the Omicron variant, our analysis did not identify a significant association between diabetes mellitus and COVID-19 severity during the Omicron-dominant period.

Research conducted in Sweden has indicated that individuals with mental illness, including dementia, are more susceptible to severe illness from COVID-19. Frailty, which is prevalent among patients with mental illness, further exacerbates the likelihood of severe illness [41]. However, the current findings revealed a significant association between dementia and severe COVID-19 only during the Omicron-dominant period. This observation may be attributed to demographic differences between the Delta-dominant and Omicron-dominant periods. Specifically, during the Delta-dominant period, the proportion of elderly individuals, who are more susceptible to dementia compared to younger individuals, was substantially lower than during the Omicron-dominant period. As dementia is more prevalent among older age groups, the relatively smaller representation of older individuals during the Delta-dominant period may have contributed to the lack of a significant association between dementia and severe COVID-19 during that period.

Our study revealed that being vaccinated with fewer than two doses was a significant risk factor only during the Delta-dominant period. This finding is consistent with other studies, which have concluded that vaccines available during Delta-dominant period appear to be effective in preventing severe complications and deaths from COVID-19 in adults of all ages [42]. In contrast, no significant effect of vaccination was found on the severity of COVID-19 during the Omicron-dominant period. Several possible explanations may account for this discrepancy. Firstly, differences in the immune response elicited by vaccines against the two variants could play a role. The Delta variant, with its higher viral load and increased transmissibility, may have been more susceptible to the immune response generated by the vaccines. Conversely, the Omicron variant, with multiple mutations in its spike protein, may have developed the ability to evade the immune response generated by the vaccines, resulting in lower vaccine effectiveness against this variant [43,44]. Secondly, vaccination coverage rates and waning immunity over time may also contribute to the observed differences. During the Delta-dominant period in Japan (1 August–30 November 2021), the second dose vaccination coverage in Hiroshima ranged from 28.97% to 70.26% [45], with the effect of the second dose likely still present among patients. However, during the Omicron-dominant period (22 December 2021–19 April 2022), the second dose of the vaccine may have waned, particularly against a new variant like Omicron. Therefore, patients admitted to hospital during the Omicron-dominant period may have experienced declining immunity, making them more susceptible to complication of infections. In addition, although the third dose of vaccination started on 1 December 2021, with healthcare workers receiving priority, followed by over-65-year-old people, and reaching a coverage rate of 46.20% by 19 April 2022 [45], sufficient time may not have elapsed for the third dose to confer protective immunity among patients admitted during the Omicron-dominant period.

Our study revealed that being vaccinated with fewer than two doses was a significant risk factor only during the Delta-dominant period. This finding is consistent with the conclusions of other researchers, who have noted that vaccines available during the Delta-dominant period appeared to be effective in preventing severe complications and deaths from COVID-19 in adults of all ages [42]. However, in contrast, our study found no significant effect of vaccination on the severity of COVID-19 during the Omicron-dominant period.

Several possible explanations may account for this discrepancy. Firstly, differences in the immune response elicited by vaccines against the two variants could play a role. The Delta variant, with its higher viral load and increased transmissibility, may have been more susceptible to the immune response generated by the vaccines. Conversely, the Omicron variant, with multiple mutations in its spike protein, may have developed the ability to evade the immune response generated by the vaccines, resulting in lower vaccine effectiveness against this variant [43,44].

Secondly, vaccination coverage rates and waning immunity over time may also contribute to the observed differences. During the Delta-dominant period in Japan (August 1–30 November 2021), the second dose vaccination coverage ranged from 28.97% to 70.26% [45], with the effect of the second dose likely still present among patients. However, during the Omicron-dominant period (22 December 2021–19 April 2022), although the third dose vaccination campaign began on 1 December 2021, and reached a coverage rate of 46.20% by the end of the study period [45], sufficient time may not have elapsed for the third dose to confer protective immunity among patients admitted during this period.

Furthermore, differences in the definition of severity and the type of vaccines administered may also contribute to discrepancies between our study and others. For example, Feikin et al. summarized vaccine effectiveness against severe Omicron diseases using hospital admission as a measure of severity [46], while another study reported a high risk of severe COVID-19 among individuals who had received fewer than two doses of any type of vaccine [15]. These differences in methodology may limit the comparability of our study with others.

This study presents several strengths and limitations that are important to consider in interpreting the findings. Regarding strength, this is the first study to evaluate risk factors of severity of COVID-19 among Japanese people during both the Delta-and Omicron-dominant periods.

The evaluation of risk factors, notably pre-existing conditions such as diabetes and hypertension, was conducted by attending physicians and documented when treatment was warranted. It is worth noting that the diagnosis of these conditions was based on clinical judgment and recorded during patient care. Occasional increases in blood pressure or high blood glucose levels that did not meet diagnostic criteria for hypertension or diabetes were not considered as such in our analysis. This approach ensures that only medically significant instances of these conditions were included, enhancing the accuracy and clinical reliability of risk factors identified in our study.

Regarding limitations, the inclusion of only 61.0% of patients hospitalized in Hiroshima Prefecture during the Delta-dominant period and 51.0% during the Omicron-dominant period may limit the generalizability of the findings. The results may not fully reflect the situation in either Hiroshima Prefecture or Japan. Furthermore, the combination of three levels of severity (mild, moderate, and very severe) into a single category due to a lack of data for each severity categorization limits the ability to differentiate the association of risk factors across different levels of severity.

The findings of this study may assist healthcare providers and physicians in prioritizing patients for hospital admissions and preventing complications from COVID-19. Awareness of changing risk factors with the emergence of new variants is crucial for effective patient management. While this study assessed several risk factors, there are likely additional factors that warrant investigation. Future research should explore a broader range of potential risk factors to provide a more comprehensive understanding of COVID-19 severity determinants.

## 5. Conclusions

The current study has identified potential risk factors for the severity of COVID-19 during both the Delta- and Omicron-dominant periods. The current findings suggest that these risk factors may not remain consistent across different variants of the SARS-CoV-2 virus. Additionally, the probability of COVID-19 severity increased with the number of identified risk factors.

These findings have significant implications for policy makers, public health specialists, and clinicians. By understanding how risk factors for COVID-19 severity vary between different variant-dominant periods, stakeholders can better prioritize screening, treatment, and non-pharmaceutical interventions. This is especially crucial in situations where there may be a shortage of healthcare resources, such as hospital beds.

Overall, our study underscores the importance of a dynamic and responsive approach to managing disease severity in the face of changing viral variants in future pandemic.

## Figures and Tables

**Figure 1 ijerph-21-00322-f001:**
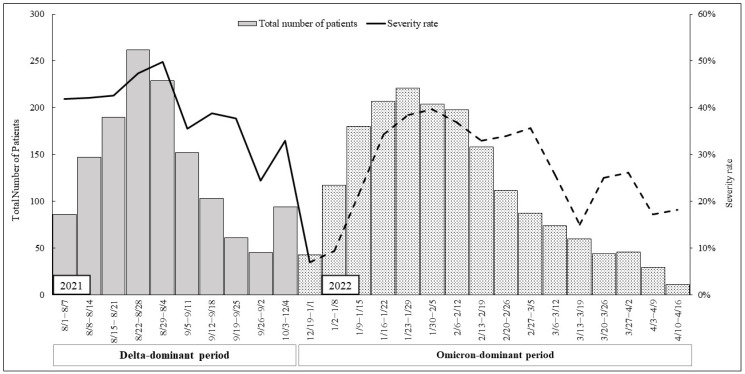
Total number of patients and severity rate during “Delta-dominant period” and “Omicron-dominant period”. The total number of patients is represented by a bar graph on the left *y* axis; the severity rate (%) is represented by a line graph on the right *y* axis.

**Table 1 ijerph-21-00322-t001:** Frequency of risk factors and severity of COVID-19.

Risk Factors	Delta-Dominant Period			Omicron-Dominant Period		
	Number of Patients	Number of Severe Cases	Severity Rate (%)	Number of Patients	Number of Severe Cases	Severity Rate (%)
Total	1367	576	42%	1790	542	30%
Age						
≥65	183	98	54%	1000	433	43%
<65	1184	478	40%	790	109	14%
Sex						
Male	803	371	46%	929	320	34%
Female non-pregnant	535	200	37%	798	220	28%
Female pregnant	29	5	17%	63	2	3%
Current smoker						
Yes	282	128	45%	138	32	23%
No	1085	448	41%	1652	510	31%
Obese (BMI > 25)						
Yes	496	266	54%	399	105	26%
No	871	310	36%	1391	437	31%
Blood pressure						
Yes	245	125	51%	674	279	41%
No	1122	451	40%	1116	263	24%
Diabetes						
Yes	155	97	63%	354	136	38%
No	1212	479	40%	1436	406	28%
Hemodialisys						
Yes	8	4	50%	49	21	43%
No	1359	572	42%	1741	521	30%
Dimentia/mental disorder						
Yes	56	27	48%	318	166	52%
No	1311	549	42%	1472	376	26%
Vaccine < 2						
Yes	1259	538	43%	725	206	28%
No	106	38	36%	1065	336	32%
Number of risk						
≤1	663	216	33%	612	80	13%
2	393	183	47%	457	148	32%
3	210	120	57%	424	178	42%
≥4	101	57	56%	297	136	46%

**Table 2 ijerph-21-00322-t002:** Association between potential risk factors and severity of COVID-19.

Risk Factor	Delta-Dominant Period				Omicron-Dominant Period			
	Odds Ratio	95% LI	95% UI	*p* Value	Odds Ratio	95% LI	95% UI	*p* Value
Age > 65	2.59	1.75	3.84	<0.0001	3.89	2.95	5.12	<0.0001
Sex (Male)	1.42	1.12	1.81	0.0039	1.76	1.40	2.21	<0.0001
Current smoker	1.08	0.81	1.43	0.6044	0.91	0.58	1.42	0.6856
Obesity	1.99	1.57	2.52	<0.0001	1.15	0.87	1.52	0.3399
Blood pressure	1.06	0.77	1.46	0.7326	1.30	1.02	1.65	0.0338
Diabetes	2.03	1.40	2.95	0.0002	1.14	0.87	1.50	0.332
Hemodialisys	1.39	0.33	5.90	0.6546	1.19	0.64	2.19	0.5853
Mental disorder	1.11	0.62	2.00	0.7298	2.22	1.69	2.92	<0.0001
Vaccine < 2 dose	2.39	1.46	3.91	0.0005	1.24	0.98	1.56	0.0684
Risk 2 ^1^	1.80	1.40	2.33	<0.0001	3.19	2.35	4.33	<0.0001
Risk 3 ^1^	2.76	2.01	3.79	<0.0001	4.81	3.55	6.52	<0.0001
Risk 4 or more ^1^	2.68	1.75	4.10	<0.0001	5.62	4.05	7.79	<0.0001

LI—lower interval; UI—upper interval. ^1^ Reference value is “zero and one risk factor”. Probability modeled is “being severe”.

## Data Availability

In accordance with the Ethical Committee protocol, the data must be kept at the Department of Public Health and Health Policy, Hiroshima University, Japan, and are not to be shared without permission of Hiroshima Prefecture, Japan. Therefore, the data shall only be made available with permission of Hiroshima Prefecture for reasonable purposes. Please initiate contact through tkubo@hiroshima-u.ac.jp to request the permission.

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
