# Peer review of "Potential Risk Factors to COVID-19 Severity: Comparison of SARS-CoV-2 Delta- and Omicron-Dominant Periods"

_ijerph, 2024, doi:10.3390/ijerph21030322_

Round 1
Reviewer 1 Report
Comments and Suggestions for Authors
The study does not provide new results regarding the risk factors for the severity of COVID-19. Those factors, described here, have been widely reported worldwide. However, it has a local importance, considering that there is not much information about it in this area.
The quality of the language must be improved.
The methodology has to describe what criteria they used to include variables into the logistic regression analysis.
In the results section, in Table 1 it is recommended to indicate the p values.
It is import to mention the date on which vaccination began in the country and the vaccination coverage in each period. Discuss the influence of vaccination in the two periods.

Comments on the Quality of English Language
The quality of the English must be improved.
Author Response
07 March 2024
MS ID#: ijerph-2879702
MS TITLE: Potential risk factors to COVID-19 severity: Comparison of SARS-COV-2
Delta- and Omicron-dominant periods
Reviewer #1:
Point-by-Point Response Letter
Thank you very much for your valuable feedback, corrections and time spent reviewing our manuscript. We have carefully considered each of your points and incorporated them into the revised version of the manuscript. As per your suggestion, we had an extensive English editing. In response to comments, we have provided both marked (track change) and clean copies. However, due to the extensive English editing, the marked copy may appear crowded and make it difficult to identify the main revisions made in accordance with Reviewer’s suggestion. Therefore, in the clean copy, modifications are indicated in red text to facilitate clarity.
Comment 1: The study does not provide new results regarding the risk factors for the severity of COVID-19. Those factors, described here, have been widely reported worldwide. However, it has a local importance, considering that there is not much information about it in this area.
Our response: We acknowledge your concern regarding the novelty of the results presented in our study. While it is true that the risk factors we discussed have been extensively reported on a global scale, we believe there are two reasons worth reporting our findings: i) Examining the impact of these factors during both the Delta- and Omicron-dominant periods among specific population may provide valuable insights into how the dynamics of COVID-19 severity may have shifted over time; ii) While the general risk factors may be well-known, our study provides local data that can inform targeted interventions and public health strategies tailored to our specific population. We have incorporated this explanation into the manuscript (Lines 48-53).
Comment 2: The quality of the language must be improved.
Our response: We initially underwent English editing for the initial submission. However, in response to the reviewer's feedback, we conducted further extensive English editing. Please refer to the marked copy for confirmation.
Comment 3: The methodology has to describe what criteria they used to include variables into the logistic regression analysis.
Our response: We have provided additional detail regarding the selection of variables in the model as per suggestion (Line 120-136).
Comment 4: In the results section, in Table 1 it is recommended to indicate the p values.
Our response: We have followed the STROBE guidelines and have refrained from including p-values in Table 1, as it is meant to provide descriptive statistics of background information. According to STROBE, “inferential measures such as standard errors and confidence intervals should not be used to describe the variability of characteristics, and significance tests should be avoided in descriptive tables” in cross-sectional studies.
Please refer to the following references:
Vandenbroucke JP, von Elm E, Altman DG, et al. Strengthening the Reporting of Observational Studies in Epidemiology (STROBE): explanation and elaboration. PLoS Med. 2007;4(10):e297. doi:10.1371/journal.pmed.0040297
Skrivankova VW, Richmond RC, Woolf BAR, et al. Strengthening the reporting of observational studies in epidemiology using mendelian randomisation (STROBE-MR): explanation and elaboration. BMJ. 2021;375:n2233. Published 2021 Oct 26. doi:10.1136/bmj.n2233
Comment 5: It is import to mention the date on which vaccination began in the country and the vaccination coverage in each period. Discuss the influence of vaccination in the two periods.
Our response: We have reflected reviewer’s comment in Lines 332-354.
EOF

Reviewer 2 Report
Comments and Suggestions for Authors
see enclosed file

Comments on the Quality of English Language
In line 14: in among, I would correct with “among”
In line 20-21: “were greater risk, I woud correct in “were at greater risk”
Line 50-51: review syntax
In line 195: “the impacts of obesity were expected”, I would correct “the impact of obesity was expected”
In line 206: “risk of developing to severe disease”, I would correct “risk of developing severe disease”
Author Response
07 March 2024
MS ID#: ijerph-2879702
MS TITLE: Potential risk factors to COVID-19 severity: Comparison of SARS-COV-2
Delta- and Omicron-dominant periods
Reviewer #2:
Point-by-Point Response Letter
Thank you very much for your valuable feedback and time spent reviewing our manuscript. We have carefully considered each of your points and incorporated them into the revised version of the manuscript. In response to your suggestion, we have provided both marked (track change) and clean copies. However, due to the extensive English editing as suggested by another reviewer, the marked copy may appear crowded and make it difficult to identify the main revisions made in accordance with Reviewer’s suggestion. Therefore, in the clean copy, modifications are indicated in red text to facilitate clarity.
Comment 1: My only concern is about the clinical relevance of the findings, considering that in my opinion such findings are not destined to result in significant modifications in our approach to future possible pandemics. Nevertheless I would leave the decision to the editorial board concerning whether in spite of my considerations the paper still deserves publication
Our response: We fully acknowledge the reviewer's concern. These findings were promptly utilized to prioritize vaccination and hospital admissions for individuals with identified risks in Hiroshima during the midst of the COVID-19 pandemic in 2021 and 2022. Subsequently, we compiled the manuscript and submitted it to PLOS Global Health. Unfortunately, it was returned after 8 months with saying that journal couldn’t find reviewers. Therefore, we partly agree that the timing of current submission may have reduced practical significance. Nevertheless, we maintain that ongoing research on risk factors remains important. We have expanded on this point in the manuscript, specifically in Lines 54-67 and Lines 374-380.
Regarding Comments 2-5, we have carefully reviewed and incorporated all the suggestions into the manuscript. We have revised the entire manuscript in terms of English to improve the clarity and understandability of the content as per another reviewer.
Comment 2: A limited number of expressions don’t appear sufficiently clear, namely: In line 14: in among, I would correct with “among”
Our response: Edited in Line 15
Comment 3: In line 20-21: “were greater risk, I woud correct in “were at greater risk” Line 50-51
Our response: Edited in Line 21
Comment 4: Line 50-51: review syntax
Our response: We apologize for any confusion, as we were unsure of the intended meaning behind this comment. Nevertheless, we have made extensive revisions to our manuscript, including thorough English editing, in an effort to ensure clarity and coherence throughout. So, we believe this comment has been addressed in revised version.
In line 195: “the impacts of obesity were expected”, I would correct “the impact of obesity was expected”.
Our response: Entire paragraph has been rewritten (Line 244-254).
Comment 5: In line 206: “risk of developing to severe disease”, I would correct “risk of developing severe disease”
Our response: Edited in Line 259
EOF
